# Relevance of Notch Signaling for Bone Metabolism and Regeneration

**DOI:** 10.3390/ijms22031325

**Published:** 2021-01-29

**Authors:** Tobias M. Ballhause, Shan Jiang, Anke Baranowsky, Sabine Brandt, Peter R. Mertens, Karl-Heinz Frosch, Timur Yorgan, Johannes Keller

**Affiliations:** 1Department of Trauma and Orthopedic Surgery, University Medical Center Hamburg-Eppendorf, D-20246 Hamburg, Germany; t.ballhause@uke.de (T.M.B.); s.jiang.ext@uke.de (S.J.); a.baranowsky@uke.de (A.B.); k.frosch@uke.de (K.-H.F.); 2Clinic of Nephrology and Hypertension, Diabetes and Endocrinology, Otto-von-Guericke University, D-39120 Magdeburg, Germany; sabine.brandt@med.ovgu.de (S.B.); Peter.Mertens@med.ovgu.de (P.R.M.); 3Department of Osteology and Biomechanics, University Medical Center Hamburg-Eppendorf, D-20246 Hamburg, Germany; t.yorgan@uke.de

**Keywords:** Notch, Jagged, bone metabolism, bone regeneration, osteoblasts, osteoclasts, osteocytes

## Abstract

Notch1-4 receptors and their signaling pathways are expressed in almost all organ systems and play a pivotal role in cell fate decision by coordinating cell proliferation, differentiation and apoptosis. Differential expression and activation of Notch signaling pathways has been observed in a variety of organs and tissues under physiological and pathological conditions. Bone tissue represents a dynamic system, which is constantly remodeled throughout life. In bone, Notch receptors have been shown to control remodeling and regeneration. Numerous functions have been assigned to Notch receptors and ligands, including osteoblast differentiation and matrix mineralization, osteoclast recruitment and cell fusion and osteoblast/osteoclast progenitor cell proliferation. The expression and function of Notch1-4 in the skeleton are distinct and closely depend on the temporal expression at different differentiation stages. This review addresses the current knowledge on Notch signaling in adult bone with emphasis on metabolism, bone regeneration and degenerative skeletal disorders, as well as congenital disorders associated with mutant Notch genes. Moreover, the crosstalk between Notch signaling and other important pathways involved in bone turnover, including Wnt/β-catenin, BMP and RANKL/OPG, are outlined.

## 1. Introduction

In 1914 John S. Dexter described notched wings in mutated *Drosophila melanogaster* [1]. In 1917 Thomas H. Morgan identified the responsible alleles and, hence, its gene was named Notch [2,3]. The Notch pathway was subsequently shown to be highly conserved during evolution. To date, four Notch receptors (Notch1-4) have been discovered in vertebrates [4]. They are expressed in almost all organs by various cell types and coordinate cell proliferation, differentiation and apoptosis, and thus cell fate decisions. Notch signals are mostly juxtracrine. One facilitated process is termed lateral inhibition that allows one cell within a group of similar cells to stand out and develop differently from its neighbors. Notch signaling is especially relevant during embryogenesis [5]. Moreover, Notch signaling was also shown to occur in the adult organism. Differential expression of Notch receptors and ligands has been observed under a variety of physiological and pathological situations, e.g., in inflammatory diseases and cancer [6,7].

In this regard, it has also been demonstrated that the Notch pathway plays a pivotal role in the development and metabolism of the skeleton [8]. Mutations in the genes encoding receptors Notch 1 to 3 all share the clinical feature of osteodysplasia. Likewise, Notch signaling was reported to influence bone remodeling and regenerative processes after fracture [9,10]. Notch receptors are upregulated during fracture healing, and their pharmacological inhibition impairs calvarial bone healing [9]. Similar effects have been reported for the Notch ligand Jagged-1 [10]. While its impact on bone development has been summarized in detail before, this review aims to provide a thorough overview on Notch signaling in bone metabolism and pathophysiology.

## 2. Bone Remodeling

Bone tissue is constantly remodeled, not only to preserve skeletal mass, structure and quality, but also to maintain homeostasis of blood calcium and phosphate levels throughout life. Bone remodeling, alternatively called bone metabolism or turnover, comprises of the two highly specialized processes bone formation and bone resorption, which are executed by the coordinated activities of the primary bone forming cells osteoblasts and osteoclasts, respectively [11,12,13]. Osteoblasts differentiate from mesenchymal stem cells (MSC) and aggregate along bone surfaces to synthesize an organic matrix consisting of collagens and other proteins that is termed osteoid [14,15]. The subsequent deposition of calcium phosphate crystals within the osteoid results in the formation of mineralized mature bone tissue. Some osteoblasts are embedded within the matrix and terminally differentiate into osteocytes. In contrast, osteoclasts derive from the hematopoietic monocyte/macrophage cell lineage. Through the influence of key cytokines, osteoclast precursor cells undergo cell fusion to form multinucleated osteoclasts that are capable of bone resorption [16] (Figure 1). Osteoclastogenesis and osteoclast function are regulated my multiple factors, among which the RANKL/OPG axis plays a major role. Both RANKL and OPG are secreted by osteoblasts. Whereas RANKL binds to RANK expressed on the osteoclast precursor to stimulate osteoclastogenesis, OPG functions as a decoy receptor and inhibits RANKL from binding to RANK, thus impairing bone resorption [17].

## 3. Structure of the Notch Receptors and Their Ligands

In humans and mice, four Notch receptors (Notch1-4) have been identified. All four Notch receptors are similar in their structure. As transmembrane receptors, they consist of an extracellular (NECD), a transmembrane and an intracellular domain (NICD). The NECD is expressed on the cell’s surface after transGolgi preprocessing. It has an N-terminus consisting of epidermal growth factor (EGF)-like repeats. The number of EGF-like repeats differs between Notch receptors (Figure 2). This main component of the NECD is followed by three Lin-12-Notch (LNR) repeats. These are, in turn, followed by the heterodimerization domain that links the NECD to the transmembrane domain. The transmembrane domain connects the NECD with the NICD. The NICD consists of a recombination signal binding protein kappa-J region associated module (RAM), followed by seven repeating Ankyrin (ANK) domains. Notch1 and -2 have a transcriptional activation domain (TAD box), which does not exist in receptors Notch3 and -4. All four Notch receptors carry a PEST domain towards their C-terminus. The PEST domain is named after its components: proline, glutamine, serin and threonine, which determines the half-life of the protein and hence time of active transcription of the NICD [18]. The PEST region is targeted by ubiquitin ligases for proteasomal degradation [8,19].

## 4. Notch Signaling Pathway

Notch ligands may be classified into two groups: canonical and noncanonical ligands. Canonical ligands mediate signaling through direct cell-to-cell signaling. One cell expresses the Notch receptor and exposes it at its membrane, and the membrane-bound ligand of another cell may associate with it (Figure 3). Thus, juxtracrine signaling is achieved. Five membrane-bound ligands have been identified for human Notch receptors: Delta-like (Dll)-1/3/4 and Jagged1 and -2. Principally, all five ligands are able to bind to all four Notch receptors [22]. Despite their very similar structure, Notch receptors do not seem to be redundant [23]. Therefore, various receptor-ligand combinations are possible and specifically transmit signals. For example, Notch activation by Jagged1 has a proangiogenic effect due to an inhibition of vascular endothelial growth factor receptor 1 (VEGFR1) expression. In contrast, Dll4 induces the expression of the antiangiogenic VEGFR1 [24]. Dll3 functions rather as a negative regulator than an activating ligand, and suppresses Notch signaling in a cell-autonomous manner [25].

Notch receptors bind their ligands mainly through the EGF repeats present in both the NECD and the ligands. Consequently, other proteins with EGF repeats can interact with Notch receptors as well. Indeed, several of these so called noncanonical ligands have been discovered [26]. This very heterogeneous group of noncanonical ligands enables paracrine cell regulation [4]. Additionally, interaction of Notch signals with Wnt-, hypoxia inducible factor (Hif)-1α, bone morphogenetic protein (BMP) and TGF-β pathways has been described [27,28].

After binding of a canonical ligand to the receptor, its cleavage site is revealed for a disintegrin and metalloprotease (ADAM). Following a second cleavage by the γ-secretase-complex, the NICD is released into the cytoplasm [20]. The NICD undergoes nuclear translocation and coactivates the expression of target genes together with the proteins Mastermind-Like 1-3 (MAML1-3), Recombination Signal Binding Protein Kappa J Region (RBP-Jκ) and coactivators [29]. Classical canonical Notch target genes are hairy enhancer of the split-1 (Hes1) and hairy ears, Y-linked (Hey), which play a pivotal role during embryogenesis [30,31].

A second pathway within the receiving cell exists. After receptor cleavage by the ADAM protease, the Notch extracellular truncation (NEXT) is internalized into the cytoplasm in form of an endosome or multivesicular body. Then, the endocytosed Notch receptor can either be further activated through the γ-secretase-complex, or it can be degraded by lysosomes. When the γ-secretase-complex has cleaved off the NICD, it undergoes nuclear translocation, as described above. The activation of endocytosed Notch receptors is suspected to be regulated by noncanonical Notch ligands [24].

## 5. The Role of Notch Signaling in Adult Bone Metabolism

Bone growth starts during embryogenesis and continues throughout childhood and adolescence. Even after final skeletal growth and maturation has been reached, bone tissue is far from being inactive. Constant bone remodeling takes place to meet biomechanical requirements. The Notch signaling pathway plays a dimorphic role in bone turnover, which is cell context-dependent. On the one hand, impaired osteoblast differentiation causing osteopenia has been observed when Notch signaling is specifically activated in immature osteoblasts [32]. In contrast, when NICD1 is overexpressed in osteocytes, bone resorption decreases, resulting in net accumulation of bone mass. On the other hand, suppression of Notch signaling in the myeloid lineage lowers bone resorption by inhibiting osteoclasts [32]. Furthermore, Notch receptors and their ligands are differentially expressed in the various types of bone cells, exhibiting different functions in osteoblasts, osteoclasts and osteocytes. Since the distinct Notch receptors fulfill different tasks in bone metabolism, the individual receptors are described separately in the following paragraphs.

### 5.1. Function of Receptor Notch1 for Bone Metabolism

The role of receptor Notch1 for bone metabolism has been controversially discussed in the past. Overexpression of Notch1 NCID in ST-2 stromal cells impairs osteoblast differentiation and mineralization by suppressing Wnt/beta-catenin signaling [33,34]. Nobta et al. noticed that inhibiting Notch1 signaling in C2C12 cells, using specific inhibitors and small interference RNA, decreased alkaline phosphatase activity as well as the mRNA expression of Bone γ-carboxyglutamate protein (Bglap), Runt-related transcription factor 2 (Runx2) and Collagen1-α1 (Col1a1) [35]. In a gain-of-function model, transgenic mice overexpressing Notch1 NICD under the control of the 2.3-kb type I collagen promoter *(Col1a1)* stimulated the proliferation of early osteoblast precursors and resulted in severe osteosclerosis, characterized by increased bone mass due to increased bone formation, but with immature woven bone [36]. Zanotti et al. created a similar NICD overexpression system in osteoblast lineage cells, but under the control of the 3.6-kb instead of the 2.3-kb collagen type I promoter [37]. The authors noticed that NICD overexpression under control of the 3.6-kb promoter resulted in lower bone volume and trabecular structure, which is an almost opposite phenotype compared to the one reported in the study by Engin et al. It was speculated that the difference may be explained by the different promoter activation and responsive elements, which seem to activate the NICD at different stages of osteoblastic cell maturation. Hypothetically, the 2.3-kb promoter is more specific to osteoblasts, allowing osteoprogenitors to proliferate and suppress terminal differentiation and mineralization. In contrast, the 3.6-kb promoter seems to repress osteoblastogenesis at an early stage. A consensus for the dispute was found in a study by Canalis et al. *2013* that employed a whole set of promoters [32]. The authors crossed RosaNotch1 mice, in which a locus of X-over P1 (loxP)-flanked STOP cassette had been placed in between the Rosa26 promoter and the NICD coding sequence, with transgenics expressing the Cre recombinase at various stages of osteoblast differentiation. This strategy allowed the preferential expression of the Notch1 NICD in either undifferentiated cells of the osteoblastic lineage (Osterix-Cre), differentiated osteoblasts (Osteocalcin-Cre), osteocytes (Dentin Matrix Acid Phosphoprotein-1-Cre) or osteoblasts and osteocytes (Col2.3-Cre). Impaired osteoblastic cell differentiation with osteopenia was observed in mice with NICD1 overexpression in both undifferentiated and mature osteoblastic cells, but only NICD1 expressed in undifferentiated osteoblasts resulted in a higher number of osteoblasts. The phenotype of mice with specific NICD1 expression in osteocytes, which represent terminally differentiated osteoblasts embedded in the bone matrix, revealed increased bone mass and a tendency towards increased bone remodeling. These data indicate a stage-dependent effect of Notch1 signaling on cells of the osteoblastic lineage. Meanwhile, the authors reported that the expression of NICD1 resulted in an initial suppression of bone resorption and increased bone volume in three-months old mice. Similarly, inhibitory effects of Notch1 on osteoblast differentiation by direct repression of Runx2 function have been reported using conditional Notch1-NICD overexpression under the control of the *Col2.3* promoter in mice [36]. It has to be noted that only the intracellular domain of receptor Notch1 was used. Thus, conclusions with regard to the functionality of the full receptor Notch1 may not be drawn. Bai and coworkers reported that osteoclastogenesis is promoted indirectly when osteoclasts are cocultured with Notch1-deficient osteoblasts. In comparison to a control group of wildtype osteoblasts, the OPG/RANKL ratio was decreased in Notch1-deficient osteoblasts. Thus, Notch1 signaling may suppress osteoclast differentiation [38]. In turn, osteoclast number and eroded bone surface are modestly reduced in female mice with conditional activation of NICD1 in differentiated osteoblasts [32].

### 5.2. Function of Receptor Notch2 for Bone Metabolism

Notch2 was reported to exhibit a stimulatory effect on osteoclastogenesis. RANKL induces Notch2 expression during osteoclast differentiation, and overexpression of NICD2 interacting with NF-κB results in activation of NFATc1, a master transcription factor regulating osteoclastogenesis [39,40,41]. Furthermore, a mutant receptor Notch2 mouse model was shown to reproduce the Hajdu Cheney syndrome (HCS). This rare genetic disorder is caused by a Notch2 gain-of-function mutation with clinical signs of acro-osteolysis [42]. The Notch2 gain-of-function model revealed sustained osteoclast activity, elevated number of osteoclasts and increased bone resorption, resulting in marked osteopenia, while osteoblast differentiation and function were not affected. In contrast, in a separate HCS mouse model, a high bone turnover was found due to increased osteoclast and osteoblast activity [43]. In yet another mouse model with cell-type specific inactivation of Notch2 in osteoblasts, increased trabecular bone mass in the proximal femur and distal tibia was observed in vivo accompanied by an increased osteogenic capacity in vitro [44]. In contrast, no skeletal phenotype was observed when Notch2 was inactivated specifically in cells of the osteoclast lineage. In Notch2 gain-of-function mutants, Canalis et al. *2017* showed that an antiNotch2 antibody targeting the negative regulatory region (NRR), previously shown to neutralize Notch2 activity, normalized bone resorption and osteopenia [45]. Moreover, the authors noticed that Notch2 gain-of-function promotes osteoclastogenesis in vitro, which was also blunted when an antiNotch2 NRR antibody was applied. Interestingly, the effect of the antibody was not observed on osteoclasts from wildtype mice. A potential explanation for this difference could be the modest basal expression level of Notch2 in osteoclast precursors, which is also in accord with the absence of a skeletal phenotype when inactivating Notch2 in the osteoclast lineage [44].

### 5.3. Function of Receptor Notch3 and -4 for Bone Metabolism

Until now only few studies on receptor Notch3 functions for bone metabolism have been published. Receptor Notch3 is expressed by bone marrow macrophages throughout the osteoclast differentiation process promoted by RANKL and M-CSF [46]. Notch3 deletion in bone marrow macrophages slightly enhances osteoclastogenesis in vitro [38]. Notch3 was also reported to inhibit osteogenesis. Transfection of siRNA targeting Notch3 in jawbone marrow-derived cells significantly increased mRNA expression of Integrin Binding Sialoprotein, Activating Transcription Factor 4 (ATF4) and Nuclear Factor of Activated T-Cells, cytoplasmic 1 (NFATc1) [47]. ATF4 is an osteogenic transcription factor fulfilling a role in terminal osteoblast differentiation. Strong evidences exist that ATF4 regulates amino acid import and promotes matrix mineralization [48,49]. ShRNA mediated silencing of receptor Notch3 upregulated expression of alkaline phosphatase (ALP), BGLAP and Runx2 resulting in an enhanced osteogenesis in primary rat bone marrow stem cells (BMSC) [46].

As for Notch4, Bagheri et al. reported a transient upregulation of Notch4 on days three and seven of osteogenic differentiation in human MSCs. At later stages of differentiation including days 14, 21 and 28, Notch4 was found to be expressed at steady levels [50]. These findings are in line with Chakravorty et al., who showed increased expression of Notch4 during osteogenic differentiation in human alveolar bone-derived osteoprogenitor cells [51]. However, due to the low expression levels of receptor Notch4 in bone cells, only few studies on that particular receptor in skeletal tissue have been performed.

Based on the above observations, the effects of Notch signaling on osteoblasts and its progenitors are not simply either stimulatory or inhibitory, but cell context-dependent. It seems that the effect of Notch signaling depends on the developmental stage of bone formation and differentiation status of each cell. A restricted activation of Notch signaling in osteoblasts leads to increased bone formation [52]. Furthermore, it seems to be essential that Notch signals are intermittent [53].

## 6. The Role of Notch Signaling in Bone Regeneration

Apart from the delicate balance between bone formation and resorption in normal bone metabolism, bone tissue exhibits a high regenerative capacity, which becomes obvious during fracture healing. Bone regeneration is a complex process, involving a series of different cell types and signaling pathways [55,56,57]. Notch signaling is considered to be of critical importance for bone regeneration. During both intramembranous and endochondral ossification, the expression of all receptors of the Notch family and respective target genes, including *Hes1*, *Hey1* and *Hey2,* were reported to be induced [58]. Especially, Hes1 is an intracellular determinant of bone mass and structure [59]. In mesenchymal progenitor cells, Notch signaling was found to be downregulated at the early stages of fracture healing [60]. Notch signaling in BMSC is critical for bone regeneration. Removal of Notch signaling in BMSC by deleting RBPjκ under the control of Peroxiredoxin-1 (Prx1) enhancer results in impaired fracture healing, but regeneration is not impaired in mice with a specific deletion of RBPjκ in osteoblasts or chondrocytes [61]. Another study reported on the fracture healing process in Mx1Cre;dnMAML mice with impaired RBPjκ-mediated canonical Notch signaling. Reduced expression of RBPjκ- signaling prolonged the inflammatory phase and disturbed chondrogenesis at 10 days post injury, while the early stages of bone regeneration were not affected [62]. Interestingly, when Notch signaling was inhibited by systemic application of the γ-secretase inhibitor N-[N-(3,5-difluorophenacetyl)-L-alanyl]-S-phenylglycine t-butyl ester (DAPT), cartilage and bone callus formation was increased via the promotion of MSC differentiation, resulting in accelerated fracture healing [63]. Moreover, DAPT application in fractures resulted in a lower OPG/RANKL ratio and higher osteoclast surface per bone surface, indicating enhanced osteoclastogenesis and bone remodeling. Notch1 signaling was also reported to positively affect the fracture healing process [10]. An inducible mouse model overexpressing NICD1 in mesenchymal progenitor cells exhibited less cartilage, more mineralized callus tissue and stronger and stiffer bones after fracture repair. Finally, injection of an antibody against receptor Notch1’s NRR reduced callus bone mass and biomechanical strength in the early stages of fracture healing.

Despite a growing number of studies on the role of Notch signaling in bone metabolism, the precise mechanisms through which Notch signaling controls bone regenerative processes are still elusive. This may, in part, be explained by the complex cell interactions and the involvement of multiple cell types in fracture healing. Moreover, due to the key role of receptors Notch1 and -2 signaling in embryogenesis and embryonic skeleton development, there are no animal models without preexisting musculoskeletal phenotypes for these receptors.

## 7. Crosstalk of Notch Signaling with Other Pathways

Bone tissue undergoes constant remodeling to adapt physical strain. Thus, a delicate functional balance of the participating cell types within skeletal tissue is required. Multiple signaling pathways work cooperatively to control the number and activity of osteoblasts, osteocytes and osteoclasts, and hence to regulate bone formation, turnover and fracture healing. Thus, it is not surprising that Notch signaling crosstalks with several important pathways including Wnt/β-Catenin, BMP, RANKL/OPG, hypoxia and Hedgehog, all of which are known to critically influence bone tissue [64] (Figure 4).

Bearing in mind that osteocytes have a longer lifespan than osteoblasts, their phenotypic regulation is of utmost importance for the integrity and maintenance of bone tissue [65]. Shao et al. reported a higher expression of *Hey1* in mature osteocytes compared to osteoblasts. Blockade of Notch signaling via DAPT led to a dysregulation of the osteocyte phenotypes with spontaneous calcium phosphate deposition and collagen fibril accumulation. The intracellular mineral transport was found to be disturbed in osteocytes [66]. In their subsequent article, the authors reported a regulation of E11 expression through *Hes1* activity. Furthermore, they found a costimulatory effect with the Wnt pathway [67]. E11, also referred to as gp38, is the earliest osteocyte-specific protein expressed as osteoblasts differentiate [68].

Findings support the idea, that Notch signaling has a time-dependent function on osteoblasts, depending on the stadium of differentiation [53]. In preosteoblasts, mTORC1 was shown to inhibit Runx2 expression by the activation of Notch signaling, which was initiated through a STAT3/p63/Jagged cascade [69]. Notch was reported to be positively regulated by mTORC1 in osteoblasts. Huang et al. observed a decline in mTORC1 activity during differentiation of preosteoblasts, yet osteoblast differentiation was enhanced after inhibition of mTORC1.

BMP is part of the transforming growth factor-β (TGF-β) family. More than 14 BMPs have been described in humans [70]. When BMPs bind to their receptors, phosphorylation of receptor Smad protein (R-Smad) takes place. In the cytoplasm, R-Smads bind to coSmads (Smad4) and translocate into the nucleus to regulate the expression of target genes, thereby interacting with other signaling pathways including the Notch pathway [71]. The interaction of NICD1 with Smad3 enhances TGF-β and Notch-dependent signaling events, whereas an interaction of NICD4 with Smad3 represses TGF-β-dependent transcription [72]. BMPs have multiple biological functions and regulate cell proliferation and differentiation during development [73]. Notch signaling enhances BMP9-induced bone formation and induces ALP activity. Cao et al. found that BMP9-induced ALP activity was significantly inhibited by either a recombinant-adenovirus containing dominant-negative mutant of Notch1, or by the γ-secretase inhibitor DAPT, in a concentration-dependent manner [74]. This effect was reversible in in vitro experiments when cells were infected with a recombinant adenovirus inducing the expression of the Notch ligand Dll1. These results indicate an impact of Notch signaling on BMP9-induced osteogenic factors, especially during the early period of osteogenic differentiation of MSCs. However, it seems that differences exist regarding the impact of the different Notch receptors and ligands. Additionally, Cao et al. suggested that Notch expression influences the expression activin-like kinase-2 abundance, representing a key receptor of BMP9 [74]. Recently, a study by Seong et al. reported that BMP9 induces the expression of the Notch target gene *Hes1* [75]. In an in vitro experiment with mouse primary osteoblasts, Notch signaling was blocked by the application of LY3039478, a specific inhibitor for the γ-secretase-complex. The simulation with BMP9 led to a dosage-dependent increase in *Hes1* expression. The authors noted that BMP9-induced expression of *Hes1* was likely to be induced via activation of Sma1/5.

Canonical Wnt/β-catenin signaling acts synergistically on osteogenic differentiation by BMP9 [76]. Tu et al. generated an osteocyte-specific transgenic model with a dominant active β-catenin in osteocytes. The authors observed that activation of Wnt/β-catenin signaling in osteocytes increases bone formation, bone mineral density and cancellous bone volume [65]. β-catenin mediates transcriptional activation of the Notch ligand Jagged1 [77]. Jagged1 represents a molecular link of the two pathways [65]. Osteocyte-specific deletion of Jagged1 results in higher expression levels of the remaining Notch ligands, receptors and target genes. Another explanation for the Notch coactivation by Wnt signaling is that the NICD stabilizes cytoplasmatic β-catenin and hence prolongs Wnt signaling [78]. Additionally, Notch and β-catenin are connected via Glycogen Synthase Kinase-3 (GSK-3β). Notch inhibits GSK-3β activity, this stabilizes SNAI1 which activates β-catenin [79].

Hypoxia-inducible factor (Hif)-1α is a transcription factor regulating the cellular response to oxygen alternations. It has a role in many physiological as well as pathological processes that require neoangiogenesis [80]. Endothelial Hif-1α expression is necessary for bone formation [81]. Hypoxia signaling leads to an upregulation of the Notch ligand Dll4 [71]. Notch signaling coactivates Hif- dependent osteogenesis and angiogenesis in mice [81,82]. Hif-1α stabilizes the nuclear NICD and prolongs the transcription of Notch target genes [83]. Thus, Notch influences angiogenesis indirectly via its interaction with the Hif-1α pathway. Notch3-deficient mice showed defects in vascular smooth muscle cells [84]. Likewise, Notch3-deficiency in zebrafish results in a loss of perivascular pericytes and the fish suffered from hemorrhage and a defective blood-brain barrier [85].

## 8. Pathologic Mutations in Notch Signaling

Most Notch-associated disorders already become apparent in the fetus. Gene mutations encoding Notch receptors or Notch ligands can lead to syndromic diseases, which are characterized by distinct symptoms. Examples of syndromes are summarized in the following.

### 8.1. Adams-Oliver Syndrome

Adams-Oliver syndrome (AOS) is a rare disease with an incidence of 1 in 225,000 [86]. It was described in 1945 for the first time. In many cases a pathogenic variant of the *NOTCH1* gene is seen. In a few instances, pathogenic variants of the gene encoding the Notch ligand *DLL-4* have been identified [8]. Pathogenic variants of *NOTCH1*, causing AOS are missense mutations that alter cysteine residues within the EGF repeats of the NECD1. These alterations lead to structural changes of the NECD1 and prevent ligands from binding to the Notch1 receptor [87].

Affected patients present with cranial ossification defects and aplasia cutis congenita (ACC). Furthermore, terminal transverse limb defects are characteristic [88]. The lower extremities are more severely affected than the upper extremities. Defects range from short distal phalanges to the complete absence of fingers and toes, or even hands and feet. Cardiovascular malformations occur in 23% of cases [87]. The majority of patients exhibit no neurological deficits [89]. The life expectancy depends on the severity of the cardiovascular malformations that frequently result in pulmonary hypertension [90]. Moreover, skin defects due to ACC facilitate skin infections. Most small defects of the scalp heal well during the first few months; deeper skin defects may result in life threating bacteriemia and sepsis [91]. Treatment options are merely symptomatic with cardiac surgery for heart malformations and plastic surgery to cover deep skin defects [92,93].

### 8.2. Alagille Syndrome

The Alagille Syndrome occurs with an estimated prevalence of 1 in 70,000 newborns [65]. For the majority of the patients, a mutation in *JAG1* or *NOTCH2* genes may be found [94]. In both instances a loss of function is seen, which leads to morphological abnormality of the bile ducts and resultant liver disease in infancy. Histology reveals paucity of bile ducts [95]. The diagnosis of Alagille syndrome can be confirmed by genetic testing, e.g., sequence analysis of JAG1 and NOTCH2 [96]. The patients may have distinct facial features, ophthalmologic abnormalities and butterfly-like shaped vertebrae with decreased bone mineralization [97]. Renal abnormalities are found in 39% of the patients with Alagille syndrome [98]. Cardiac malformations with resemblance to Fallot Tetralogy have been reported. Furthermore, patients with Notch2 loss-of-function mutations have a smaller marginal zone of the spleen with fewer B-cells [99]. Treatment options are multidisciplinary depending on the affected organs. Usually, patients require a liver transplant in early childhood [100].

### 8.3. Hajdu Cheney Syndrome (HCS)

HCS is a rare skeletal disorder; less than 100 patients with HCS have been reported in the literature [101]. It is associated with a gain-of-function mutation in the gene encoding the Notch2 receptor [102]. HCS was first described as a cranioskeletal dysplasia by Hajdu and Kauntze in 1948. In 1965 Cheney observed acro-osteolysis in one case [103]. HCS follows an autosomal dominant inheritance, but many cases occur due to spontaneous genetic mutation. Patients suffer from severe osteoporosis and pathogenic acro-osteolysis [104]. Furthermore, the skeleton is characterized by short stature, vertebral abnormalities, long bone deformities and joint hypermobility [101]. The experimental murine data of Vollersen et al. and Zanotti et al. suggested increased osteoclastogenesis, which leads to higher bone turnover [43,105]. The enhanced osteoclastogenesis has been attributed to increased and sustained Notch signaling in osteoblasts that in turn stimulates osteoclast precursor cells via modulation of the RANKL/OPG axis. Under wildtype conditions, the activated NICD2 from receptor Notch2 is degraded in the proteasome following F-box/WD repeat-containing protein 7 (FBXW7)-mediated ubiquitination. Notch2 mutations associated with HCS result in a truncated C-terminus that lacks the PEST domain and enables the NICD2 to escape from FBXW7-mediated ubiquitination and degradation. Mice with osteoclast-specific Fbxw7 ablation showed elevated Notch2 expression levels and a similar skeletal phenotype to that observed with Notch2 gain-of-function mice [106]. In turn, administration of Notch2 inhibitors rescued osteoporosis caused by FBXW7-deletion. Thus, aberrant intracellular quantities of FBXW7/NOTCH2 are one of the potential mechanisms of HCS [106].

An antibody targeting the NRR of Notch2 was able to reverse osteoporosis in a murine Notch2 HCS model [45]. Further investigation is necessary, and a translation of the findings in mouse models to human diseases might not be fully possible. A blockade of Notch signaling in humans likely results in gastrointestinal side effects, bone marrow toxicity and the development of vascular tumors [107,108,109].

Patients diagnosed with HCS are cared for by addressing the symptoms. Promising results of have been reported for denosumab [110]. Denosumab was the first biological pharmakon used in osteoporotic high-risk patients to prevent fractures. Meanwhile, Romosozumab, another biological for osteoporosis, has been approved and may minimize adverse drug effects [111].

### 8.4. Lateral Meningocele Syndrome (LMS)

LMS, an extremely rare condition, was described for the first time in 1977 and is also known as Lehman syndrome [112]. Only 14 individuals with LMS have been published in the literature [113]. A point mutation of exon 33 in the *NOTCH3* gene causes LMS [114]. The mutation is upstream of the PEST domain. This suggests that Notch3 proteins are stabilized, which leads to a NOTCH3 gain-of-function [115,116]. Additional findings are facial anomalies, hypotonia and meningocele formation causing neurological dysfunction [117]. Lateral meningoceles are very rare in general. However, they occur in higher frequency in LMS. The meningoceles herniate through the intervertebral foramina. The spinal column features scalloped vertebral bodies with subsequent scoliosis [113]. Additionally, a Chiari type 1 malformation is often found in the cervical spine. Affected patients have a developmental delay, sarcopenia, syringomyelia and display increased bone remodeling.

LMS is often associated with Neurofibromatosis type 1, although the genetic background for the coappearance of both diseases is elusive [114]. In a murine model for LMS, Yu et al. were able to reverse the skeletal phenotype (osteopenia) of LMS by applying an antibody directed against the NRR of receptor Notch3 [118].

### 8.5. CADASIL Syndrome

The CADASIL syndrome is a small vessel disease. A point mutation in the NOTCH3 gene has been identified as genetic cause for the syndrome [119]. It leads to a loss of function of NOTCH3. The prevalence is estimated between two to five per 100,000 inhabitants [120]. In a minority of CADASIL patients, mutations only occurred in the EGF repeat 10 and 11 of Notch3 [121]. However, this was found to be sufficient to disable the canonical ligands Jagged and Dll from binding to the NECD3.

Patients suffer from recurrent strokes caused by an microangiopathy of the brain-supplying arteries. Often, first symptoms include migraine attacks with an atypical aura. Usually, CADASIL occurs around the age of 30 years in patients, although its onset is highly variable. The cerebral microangiopathy subsequently leads to early-onset dementia. Besides the brain, other organs are affected too, especially organs with micro vessel architecture such as the kidney [122]. A specific bone phenotype has not been described. The only current therapeutic concept includes anticoagulation to improve rheology [123]. Life expectancy on average has been reported to be 64.6 years in men and 70.7 years in women [120].

## 9. Conclusions

Receptors of the Notch family and their signaling play an essential role during the development of the skeleton, as well as in physiology of the bone metabolism and regeneration (Figure 5). Mutations in genes encoding either the Notch receptors or their ligands lead to severe functional impairment and syndromic diseases. Most of these clinical presentations exhibit a phenotype in bone tissue and present with skeletal dysplasia or abnormalities of the extremities. Multiple in vitro and in vivo models demonstrate regulatory functions of the Notch pathway in bone cells, especially in the osteoblastic lineage. The role of Notch signaling in osteoclasts is starting to emerge. As such, a detailed understanding of Notch signaling for bone metabolism and regeneration may open the window for novel treatment approaches in skeletal disorders.

## Figures and Tables

**Figure 1 ijms-22-01325-f001:**
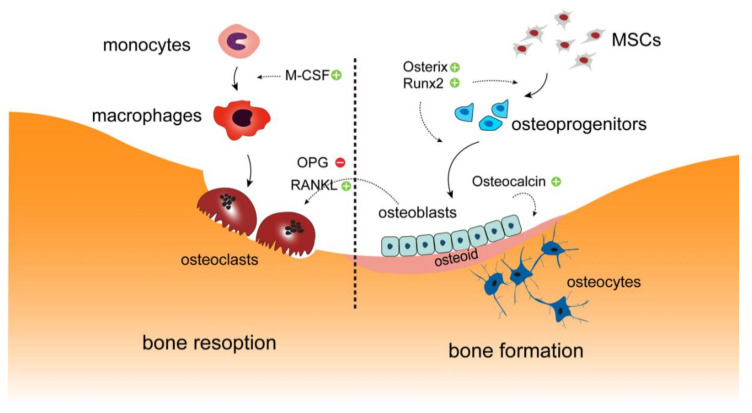
Overview on bone metabolism. Bone formation and bone resorption is carried out by osteoclast and osteoblast, respectively. In bone tissue, monocytes are recruited at or near the bone surface and differentiate into mature macrophages under the influence of M-CSF, after which they fuse to form osteoclasts. RANKL and OPG secreted by osteoblasts regulate the formation of osteoclasts. These multinucleated cells secret acid and proteolytic enzymes to break down bone tissue. Osteoblasts differentiate from mesenchymal stem cells (MSC) and aggregate along bone surfaces to synthesize osteoid. The osteoid is subsequently calcified by the integration of hydroxyapatite crystals, thus forming the mineralized bone matrix. Some osteoblasts are trapped in this newly formed bone and differentiate into osteocytes. Osterix and Runx2 are transcription factors that are essential for osteoblast differentiation and are highly expressed at both early and late stages of differentiation. Osteocalcin produced by osteoblasts is expressed during bone mineralization, which may be utilized as a marker for bone formation.

**Figure 2 ijms-22-01325-f002:**
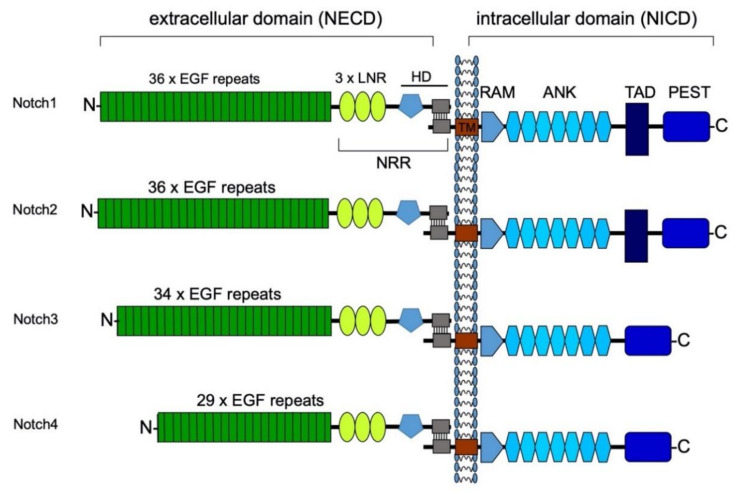
Composition of the Notch receptors. All four Notch receptors are similarly structured with an extracellular and an intracellular domain. The NECD consists of EGF repeats, which vary in their number depending on the type of Notch receptor. These are followed by three Lin-12-Notch (LNR) repeats. Together with the heterodimerization domain, this hydrophobic region forms the negative regulatory region (NRR) in direct neighborhood to the cell membrane. In this way, the NRR prevents accidental cleavage by metalloproteases [20]. A small transmembrane domain (TM) connects the extracellular domain with the NICD. The NICD bears a RAM domain towards its N-terminus. Together with the transcription factor MAML, the RAM domain forms the promoter complex, which binds to the DNA and enables transcription of Notch target genes. Furthermore, all four NICD have seven Ankyrin (ANK) domains [19]. NICD1 and two have an additional transcriptional activation domain (TAD) box, which is not present in NICD3 and 4. Towards the C-terminus, all four NICD have a PEST region. This region is named after its four most present amino acids: proline, glutamine, serin and threonine. The PEST region is the target for degradation by proteolysis [19]. Modified from Raffetseder et al. Nephron Exp Nephrol 2011 [21].

**Figure 3 ijms-22-01325-f003:**
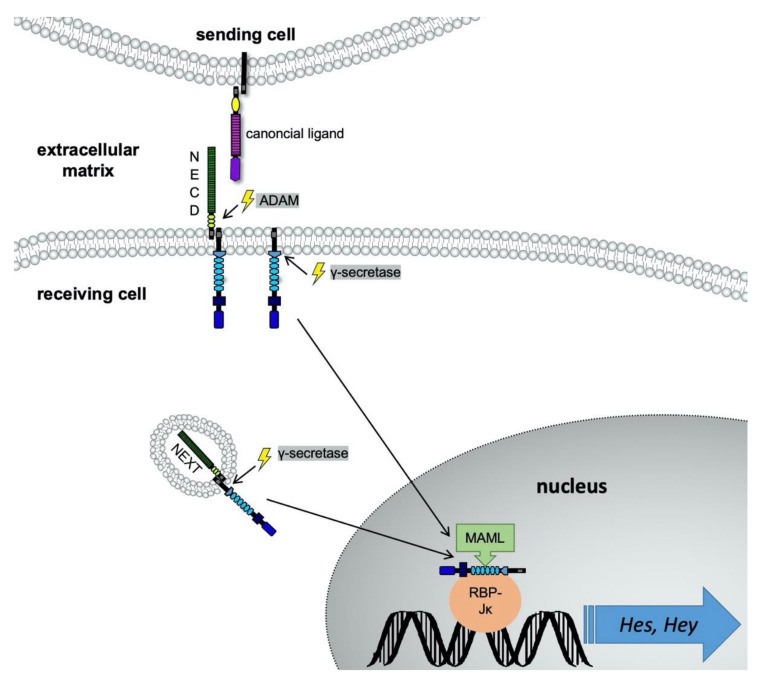
Schema on ligand binding to receptors of the Notch family. The canonical ligands of Notch receptors are expressed on the extracellular membrane. In this image, the sending cell exposes a canonical Notch ligand, e.g., Jagged1. The ligand binds to the NECD of a Notch receptor, expressed on the receiving cell’s extracellular surface. The binding takes place via protein-protein interactions of the EGF repeats from both proteins. The ligand physically pulls the receptor lengthwise, which leads to a change in protein conformation of the receptor. The extracellular cleavage site for the ADAM-proteases becomes accessible and is cleaved. This allows the intracellular domain to be cleaved off in a second step by the γ-secretase-complex. Consequently, the NICD is translocated into the nucleus. The NICD’s RAM domain forms a promoter complex with the transcription factor MAML, RBP-Jκ and coactivators. Via RBP-Jκ the complex binds to the DNA and enables regulation of Notch target genes, such as *Hes1* and *Hey1*. After cleavage of the ADAM-protease, the Notch extracellular truncation (NEXT) is internalized in an endosome. In case of activation, the NICD is cleaved off by γ-secretase-complex and then undergoes nuclear translocation. This second way of signal transduction seems to be preferred in noncanonical Notch signaling [24]. Modified from Brandt S et al. Eur J Cell Biol 2012 [54].

**Figure 4 ijms-22-01325-f004:**
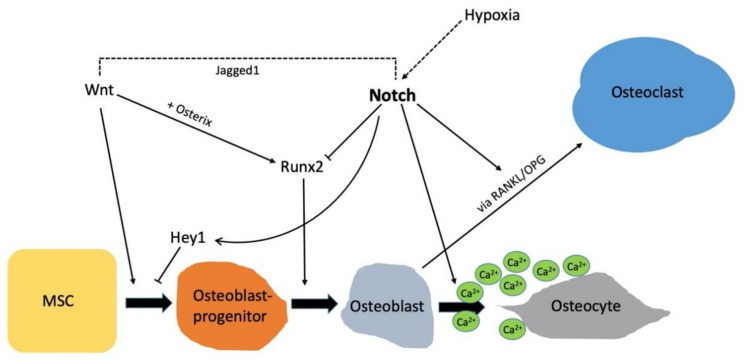
Notch-dependent signaling events and signaling crosstalk during the differentiation of MSCs towards osteocytes. Pluripotent MSCs differentiate towards osteoblast progenitors under the influence of PTH, BMP-2 and Wnt-Signaling. The Notch-regulated gene Hey1 inhibits differentiation towards osteoblast progenitors. Notch signaling is enhanced by stimulation of the Hif-1α pathway. Notch activation functions inhibitory on Runx2 and thus on further differentiation of osteoblast progenitors towards osteoblast. The osteoblast entombs itself by synthesizing minerals osteoid. The deposition of minerals and collagens is regulated by Notch [66]. Once the osteoid is mineralized, the osteoblasts are terminally differentiated and become osteocytes. Notch has an indirect influence on osteoclasts via modulation of the RANKL/OPG axis. Notch inhibits OPG, which is an inhibitor of Rank, resulting in an indirect activation of osteoclasts by Notch signaling. Arrows indicate an activation/ induction, and lines with a horizontal bar indicate inactivation/repression.

**Figure 5 ijms-22-01325-f005:**
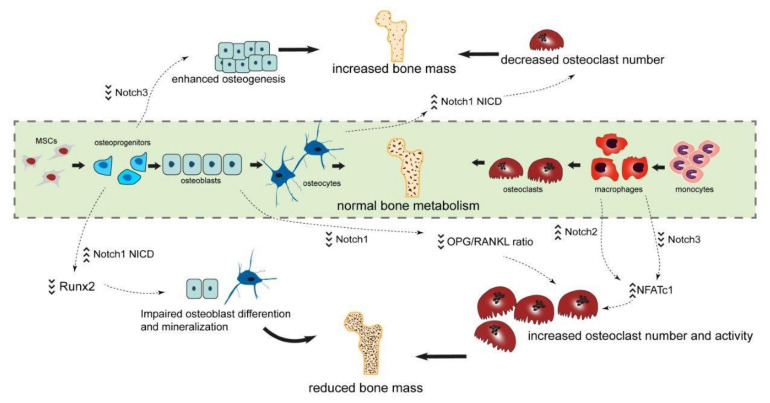
Summary of the relevance for Notch signaling for bone metabolism and regeneration. The balance of osteoblasts and osteoclasts is the key to physiological bone metabolism. Overexpression of Notch1 in osteoprogenitors impairs osteoblastic differentiation through inhibition of Runx2 and Osx, resulting in osteopenia. In contrast, overexpression of NICD1 in osteocytes causes increased bone mass, due to a decreased number of osteoclasts and reduced osteoclast function. When osteoclasts are cocultured with Notch1-deficient osteoblasts, osteoclastogenesis is promoted indirectly because of a lower OPG/RANKL ratio in comparison to wildtype osteoblasts. NICD2 interacts with NF-κB and thus activates NFATc1 transcription. This results in an elevated number of osteoclasts and increased bone resorption. Notch3 deletion in bone marrow macrophages increases mRNA expression of NFATc1 and enhances osteoclastogenesis. Silencing Notch3 in primary MSCs upregulates expression of Alp, Bglap and Runx2, resulting in an enhanced osteogenesis.Dotted arrows indicate a direct influence. Bold arrows depict an overall impact. Repeating upwards/downwards arrows indicate overexpression/reduced expression.

## Data Availability

Data is contained within the article.

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
