# Peer review of "Relevance of Notch Signaling for Bone Metabolism and Regeneration"

_ijms, 2021, doi:10.3390/ijms22031325_

Round 1

Reviewer 1 Report

A timely review covering a large number of studies on Notch signalling and bone biology. This review will be of interest to both Notch signalling aficionados and bone biologists. I have only some minor suggestions for how to improve this piece.

  • Row 123: remove “dependent”.

  • Check the space before each [Ref]. Sometimes there is a space, sometimes not.

  • Row 150: Strange sentence.

  • Row 235: Should this sentence not state that Notch signalling can be either stimulatory or inhibitory, but this depends upon context?

  • Figure 3: The key Notch signalling component RBP-Jk is missing from the figure.

  • The review covers a large number of in vivo and in vitro studies of Notch, but there is no cartoon outlining the roles of Notch signalling in bone biology. Figure 4 outlines some of the interplay with HIF and WNT, but could be expanded. Or another figure added.

Reviewer 2 Report

In this paper Ballhause et al. present a nice review of the important Notch signaling in bone metabolism and regeneration. While there is already a wealth on literature describing the Notch pathway in bone homeostasis and disease the authors give a new comprehensive view by adding new research from 2019 and 2020, as well as by including Wnt/β-catenin, BMP and RANKL/OPG. A similar review has not been done in English up to now according to PubMed.

The work is well organized, with a multitude of references to the field. The figures are well done.

English language and stlye are overall fine, however there are quite a few spelling/style mistakes: "described a notched wings" in line 34 should be changed to "described notched wings", "i-duction" in line 358 to "induction", "linage" in line 457 to "lineage" and "osteoclast" in line 457 to "osteoclasts".

Reviewer 3 Report

This interesting review highlights the role of Notch in bone metabolism and regeneration. It is well written. All the data dissect the multiple functions performed by Notch receptors in the bones.

Major points.

Line 137-148. The ligands are very briefly described. I would suggest additional information.

-Figure 3. Line 238-239. The figure partially describes the canonical Notch signaling. Additional elements, like RBPjK and coactivators, may be added also in support of what is described in the text (line 1332-134). The figure needs to be conceptually revised, also for what concerns the NOTCH-ICD, which is incorrect. There is no change after gamma-secretase cleavage which is not true (Nat Rev Drug Discov. 2020 Dec 8. doi:10.1038/s41573-020-00091-3; Expert Opin Ther Targets. 2018 Apr;22(4):331-342).

 Minor points

-Line 218. The following sentences probably require a reference that seems not to be inserted in the text. “Notch3 deletion in bone marrow macrophages slightly enhances osteoclastogenesis in vitro. Notch3 was also reported to inhibit osteogenesis. “ please insert it.

line 68 please rewrite osteoclastogenesis, ……..

-line 133 please correct into classical canonical Notch target…..….

-Please uniform Maml and MAML in the text.

Please check through text for typos.
